# *LiRo*: Benchmark and leaderboard for Romanian language tasks

**Stefan Dumitrescu**
Independent researcher

**Petru Rebeja**
Alexandru Ioan Cuza
University of Iași

**Beata Lorincz**
Technical University of Cluj-Napoca

**Mihaela Gaman**
University of Bucharest

**Andrei-Marius Avram**
Politehnica University of Bucharest

**Mihai Ilie**
Independent researcher

**Andrei Pruteanu**
Independent researcher

**Adriana Stan**
Technical University of Cluj-Napoca

**Lorena Rosia**
Deloitte

**Cristina Iacobescu**
Deloitte

**Luciana Morogan**
Military Technical Academy

**George-Andrei Dima**
Politehnica University of Bucharest

**Gabriel Marchidan**
Feel IT Services

**Traian Rebedea**
Politehnica University of Bucharest

**Madalina Chitez**
West University of Timisoara

**Dani Yogatama**
DeepMind

**Sebastian Ruder**
DeepMind

**Radu Tudor Ionescu**
University of Bucharest

**Razvan Pascanu**
DeepMind

**Viorica Patraucean**
DeepMind
viorica@google.com

## Abstract

Recent advances in NLP have been sustained by the availability of large amounts of data and standardized benchmarks, which are not available for many languages. As a small step towards addressing this, we propose *LiRo*, a platform for benchmarking models on the Romanian language on nine standard tasks: text classification, named entity recognition, machine translation, sentiment analysis, POS tagging, dependency parsing, language modelling, question-answering, and semantic textual similarity. We also include a less standard task of Romanian embeddings debiasing, to address the growing concerns related to gender bias in language models. The platform exposes per-task leaderboards populated with baseline results for each task. In addition, we create three new datasets: one from Romanian Wikipedia and two by translating the Semantic Textual Similarity (STS) benchmark and the Cross-lingual Question Answering Dataset (XQuAD) into Romanian. We believe *LiRo* will not only add to the growing body of benchmarks covering various languages, but can also enable multi-lingual research by augmenting parallel corpora, and hence is of interest for the wider NLP community. *LiRo* is available at
https://lirobenchmark.github.io/

## 1 Introduction

Recent years have seen rapid progress on many language understanding tasks, from language modelling [e.g. 4] to translation [e.g. 27] or Q&A [e.g. 21]. Most of these understandably have happened

35th Conference on Neural Information Processing Systems (NeurIPS 2021) Track on Datasets and Benchmarks.

in English, relying on the proliferation of datasets [e.g. 7, 33] and on easy access to leaderboards and benchmarks[1] [e.g. 43] that facilitate communication and standardization of experiments. Unfortunately, a similar level of access is lacking for many other languages. In this work, we focus on Romanian and aim to provide datasets and tools to facilitate research on Romanian language tasks.

Romanian is an Indo-European Romance language that evolved in relative isolation compared to other Romance languages, leading to its unique characteristics. In particular, Romanian has mixed linguistic typology [8], displaying characteristics from two different families: Romance languages [23] and Balkansprachbund [39]. For example, the majority of verb forms in Romanian function syntactically as in other languages included in the Italian branch of the Indo-European Romance language family, with the shift from Latin to Romance manifesting as the shift from synthetic/inflectional towards analytic/syntagmatic constructions (e.g. Latin *feci*, Italian *ho fatto*, Romanian *am făcut*). However, the geographical proximity to the Balkan region accounts for the existence of verb forms, such as the volo future [26], that are common to Romanian and Slavic languages (e.g. Romanian *voi face*, Bulgarian *shte napravya*). Similarly, other features, such as the enclitic definite article, attached in Romanian at the end of the noun (e.g. *omul → the man*) can either be explained through post-Roman regional contact in the Balkans or the influence of the Ancient Greek on Vulgar Latin [12]. The case of double negatives (e.g. *nu am mâncat nimic*), also present in French, Spanish and Italian, represents a challenge for ML algorithms trained on English language, where double negation are rather infrequent (e.g. *haven't eaten anything*). Furthermore, lexical similarity analyses emphasize the particularity of Romanian within the Romance group [18]. These are all arguments that support the latest findings in cross-lingual NLP studies stating that typological properties of languages impact allegedly *language-agnostic* models [20]. Hence, evaluating cross-lingual models on Romanian can contribute to shedding light on their predictive performance.

Although Romanian is spoken by around 25 million speakers, it is still considered a low-resourced language in terms of digital resources and NLP tools [40]. Within the European Language Grid [34], Romanian is listed with only 129 resources, tools and services, as opposed to English (2342), Spanish (658) or German (777).[2] To address this issue, we propose *LiRo* (**Li**mba **Ro**mână = Romanian Language), the first benchmark and leaderboard targeting models for Romanian language tasks. Currently, it includes nine standard tasks (text classification, named entity recognition, machine translation, sentiment analysis, part-of-speech tagging, dependency parsing, language modelling, question-answering, semantic textual similarity) and an extra task of gender-debiasing of Romanian language embeddings. When selecting the nine tasks in the benchmark, we aimed to have a good task alignment with benchmarks in other languages and to enable cross-lingual tasks that allow bringing the Romanian language to the attention of international researchers working on cross-lingual studies. The gender-debiasing task was included to state the importance of studying language biases in ML models [19] and to encourage research in this direction also for the Romanian language.

Along with the platform, we introduce three new datasets: RO-STS (Romanian translation of the Semantic Textual Similarity dataset [6]), XQuAD-ro (the Romanian component of the XQuAD dataset [1]), and Wiki-ro (Romanian Wiki for language modelling evaluation). For part-of-speech tagging and dependency parsing, we rely on the Romanian version of UD-RRT [2], but we propose a cross-genre training-vs-testing split in order to measure the robustness of existing systems to stylistic changes – a relevant task for Romanian language, which tends to change its form across domains.

We provide baseline results for all the tasks either by extracting results from the literature for existing datasets or by creating new baselines for the newly-created datasets and the newly-created splits. We analyse the results of the new baselines and point to directions of improvement.

## 2   Related work

One of the first initiatives for the common evaluation of disjoint Natural Language Understanding (NLU) tasks was the General Language Understanding Evaluation [GLUE; 43] benchmark. Wang et al. [43] gathered nine tasks including question answering, sentiment analysis, and textual entailment, as well as their associated training and test datasets. GLUE also includes a diagnostic dataset to analyze models' performance with respect to a wide range of linguistic phenomena found in natural language.

---

[1]This includes websites such as `paperswithcode.com`.

[2]As of June 7, 2021, in the ELG Release 2, at `https://live.european-language-grid.eu/catalogue/`.

However, the rapid advancements in deep learning led to a quick saturation of the benchmark [42] where several models surpassed non-expert humans. Wang et al. [42] proposed SuperGLUE, a novel benchmark that includes a more diverse and challenging set of tasks. Additionally, SuperGLUE can showcase significant performance gaps between BERT-like models [13] and humans.

McCann et al. [29] introduced the Natural Language Decathlon (DecaNLP), a benchmark that comprises ten NLP tasks ranging from machine translation, question answering and summarization to sentiment analysis, relation extraction and semantic parsing. Poliak et al. [31] introduced the Diverse Natural Language Inference Collection (DNC), comprising 8 tasks and 13 existing datasets. DNC is aimed at evaluating a model's capability to perform various types of reasoning. Another landmark collection of datasets for the English language was proposed by Conneau and Kiela [10]. SentEval [10] is advertised as a toolkit for the centralized evaluation of universal sentence representations. It is composed of 7 distinct tasks and 13 datasets. Different from the previous benchmarks, Evaluating Rationales And Simple English Reasoning (ERASER) [14] is a benchmark aiming to assess the interpretability of NLP models. The main contribution of this benchmark is the design of novel evaluation metrics to measure the alignment between human and model rationals. DeYoung et al. [14] establish that a rational is the evidence that supports a decision.

The aforementioned benchmarks are all based on English datasets. Recently, some effort has been dedicated to the development of multi-lingual benchmarks. XTREME [22] is a benchmark dedicated to the evaluation of cross-lingual generalization on 40 languages. Perhaps the most important observation of Hu et al. [22] is that state-of-the-art models for English exhibit sizeable performance gaps when transferred across languages. While the number of languages and the size of XTREME is remarkable, we emphasize that Romanian is not included. Through *LiRo*, we aim to establish a NLU benchmark for Romanian. Among the datasets included in the XTREME benchmark is the Cross-lingual Question Answering Dataset [XQuAD; 1] for which we provide a translation into Romanian by professional human translators, which was also added to the official XQuAD repository.

While some research works went towards creating multi-lingual benchmarks, other works focused on building mono-lingual benchmarks for understudied languages. For instance, a recently developed language-dependent benchmark is the Polish version of GLUE, known as the KLEJ benchmark [35]. KLEJ contains a set of 9 evaluation tasks for the Polish language understanding. The authors collated existing datasets together with a new dataset for sentiment analysis. The platform provides evaluation code and a public leaderboard. Another example of mono-lingual NLU evaluation is IndoNLU [44], a benchmark dedicated to the Indonesian language. IndoNLU is composed of twelve tasks. The diversity of the tasks is ensured by selecting datasets from various domains and with different styles. Recently [30] proposed KLUE as a benchmark for the Korean language, also modelled after the GLUE benchmark.

Our platform currently includes 10 tasks, 8 datasets (out of which three are new) and a public leaderboard. We pledge to further develop *LiRo* and include additional datasets and tasks to provide a comprehensive evaluation platform for Romanian and multi-lingual language tasks.

## 3   *LiRo* benchmark and leaderboard

**Benchmark.** *LiRo* is an open-source benchmark and a continuous-submission leaderboard, concentrating public Romanian datasets (existing and new) in specific tasks. The integration of datasets and tasks with model performance and efficiency allows both academia and industry to quickly gauge performance on tasks of interest. The benchmark also provides an overview of the Romanian NLU SoTA and direct access to relevant papers. Finally, it intends to foster a constructive competition and innovation by bringing together and promoting previously disparate resources.

*LiRo* is structured into *areas*, *tasks*, and *datasets*. In this paper, we focus on the NLP area, but in the future we intend to extend *LiRo* to other areas like speech or image captioning. Each area can have any number of tasks and for each task we can have any number of datasets, each with their own metric(s). *LiRo*'s homepage lists all available tasks, grouped by area. Each task contains a succinct description and the available datasets. A dataset is a specific corpus with defined training and evaluation splits, together with evaluation metrics and scripts to compute these metrics. A dataset can belong to multiple tasks—for example the Universal Dependencies Romanian RRT Treebank dataset [2] is used in POS tagging and parsing tasks. To keep things simple, *LiRo* does not host the datasets directly. Instead, we link to each individual resource's webpage while having a dedicated

| #   | Task                        | Dataset          | Metrics            | Score | Baseline   |
|-----|-----------------------------|------------------|--------------------|-------|------------|
| 1.  | Text Categorization by Topic | MOROCO          | Macro F1           | 88.03 | [17]       |
| 2.  | Named Entity Recognition    | RONEC v1.0       | Exact Match F1     | 85.88 | [15]       |
| 3.  | Machine Translation         | WMT-16-ro-en     | BLEU, ROUGE-L      | 38.5  | [28]       |
| 4.  | Sentiment Analysis          | LaRoSeDa         | F1                 | 54.30 | [38]       |
| 5.  | POS Tagging                 | UD Ro-RRT (cross) | UPOS F1, XPOS F1  | 95.73 | this paper |
| 6.  | Dependency Parsing          | UD Ro-RRT (cross) | UAS F1, LAS F1    | 88.97 | this paper |
| 7.  | Language Modelling          | Wiki-ro          | Perplexity         | 28.0  | this paper |
| 8.  | Question Answering          | XQuAD-ro         | F1, EM             | 83.56 | this paper |
| 9.  | Semantic Textual Similarity | RO-STS           | Pearson, Spearman  | 0.81  | this paper |
| 10. | Gender debiasing            | Ro embeddings    | Modified-WEAT      | 2.57  | this paper |

Table 1: Tasks, datasets, associated metrics, and baseline results available in *LiRo*. Where there is more than one metric, only the result for the first one is reported here to reduce clutter. At the moment, *LiRo* contains 10 tasks with associated datasets. The baseline results for the first 4 tasks are from the top performing models existing in the literature (and included in *LiRo*), whereas for the remaining 6 tasks, we propose new datasets or new dataset splits and associated baselines.

description page for each dataset, with statistics about the dataset, metrics, and other details useful for anyone who wishes to use them. For the newly-created datasets, we include details regarding licensing.

**Leaderboard.** Each dataset has its own leaderboard, both graphically displayed as an interactive chart, and as a table listing all participating models. For each model, we include (1) the rank of the model in the leaderboard, (2) model name, (3) metric values, (4) whether the model was trained on extra training data, (5) model size (number of parameters), (6) link to the model's paper and online code repository if any, and (7) submission date. In contrast to other benchmarks, we decided to require model size as a first step towards evaluating not only performance but also computational efficiency, following recent trends focusing on green AI [37].

We chose to have a separate leaderboard per dataset. Other platforms formulate all tasks in a common setting (e.g. convert all tasks into a binary classification [35]), so that they can provide an aggregated score. However, we found that this can lead to artificial tasks and opaque scores that might not capture the performance of the models in a meaningful way, harming understanding. Hence, we decided to create separate leaderboards and use standard problem formulations and metrics.

To submit a model to the leaderboard, we provide a templated submission form that users have to fill in. The maintainers of the platform then request additional info if needed. Once a submission is approved by a maintainer, the new model's results will be automatically displayed on the website. A similar process is used for submitting new tasks or datasets to the leaderboard.

## 3.1 Available Tasks

We list below the tasks currently included in the benchmark and their associated datasets and metrics. For a summary, see Table 1.

**1. Text Categorization by Topic**: is the task of assigning a sentence or document to an appropriate category. Currently, *LiRo* contains the MOROCO dataset [5] with a Romanian and Moldavian news classification task.

**2. Named Entity Recognition**: is the task of identifying and labeling entities in a text with their corresponding type (e.g. person, date, location, etc.). We use RONEC [16], a fine-grained dataset of 5,127 sentences annotated with 16 classes, totalling 26,376 annotated entities.

**3. Machine Translation**: is the task of translating a sentence from a source language to a different target language. Currently this task includes WMT16 RO-EN dataset [3], a classic translation corpus used in several NLP papers.

**4. Sentiment Analysis**: requires classifying the affective state of a text, most frequently labelled as positive or negative. We include the recently proposed LaRoSeDa dataset [38], the first and only public dataset to our knowledge for this task in Romanian.

**5. Part-of-Speech Tagging**: (POS tagging) is the task of tagging a word in a text with its part of speech. We use the standard Romanian dataset for this task, the Universal Dependencies Romanian RRT Treebank (UD-RRT) [2], but we propose a different train-test split of the data in order to evaluate robustness across genres (see details in Section 5). UD-RRT has annotations for Universal Parts of Speech (UPOS) as well as language-specific parts of speech (XPOS).

**6. Dependency Parsing**: is the task of extracting a dependency parse of a sentence that represents its grammatical structure and defines the relationships between "head" words and words, which modify those heads. We use again UD-RRT with the same splits as in Task 5. UD-RRT offers multiple layers of annotation for dependency parsing.

**7. Language Modeling**: is the task of predicting the next word or character in a document. For evaluating models on this task, we release the Romanian Wiki dataset, described in the next section.

**8. Question-answering (QA)**: The task is to answer a question given a segment of text as context. As the first such dataset in Romanian, we introduce XQuAD-ro, the Romanian translation of XQuAD [1]. XQuAD follows the standard SQuAD [1, 33] setting for QA: given a context paragraph, the model has to answer questions whose answers (of variable length) are spans in the context paragraph. XQuAD-ro is further detailed in the next section.

**9. Semantic Textual Similarity**: Given a pair of sentences, this regression task measures how similar the sentences are. We introduce RO-STS as the Romanian translation of STS [6], see next section for more details on this dataset.

**10. Language embeddings debiasing**: Given the growing concern about the negative impact that gender-biased language embeddings may have in practical applications, we measure the gender bias in existing Romanian language embeddings using the method proposed in [45] for languages with grammatical gender, and invite contributors to submit debiasing methods that can lower the gender bias in existing embeddings, or submit less biased embeddings. More details in section 5.

## 4 Newly-proposed datasets

We introduce three new datasets: *RO-STS*, *XQuAD-ro*, and *Wiki-ro*. The *RO-STS* and *XQuAD-ro* datasets were carefully translated from English and are the first of their kind for Romanian. The *Wiki-ro* is the first officially published Wiki dump for the Romanian language, purposely-cleaned with the aim of standardasing language model evaluation.

**RO-STS.** The *RO-STS* (Romanian Semantic Textual Similarity) dataset is the Romanian translation of the STS English dataset[3]. RO-STS contains 8,628 sentence pairs with their similarity scores. The original English sentences were collected from news headlines, captions of images and user forums, and are categorized accordingly. The Romanian release follows this categorization and provides the same train/validation/test split with 5,749/1,500/1,379 sentence pairs in each subset. Using both translations and similarity scores, *RO-STS* can be used for (at least) two purposes: (1) as a textual similarity dataset for Romanian, and (2) as a parallel Romanian-English dataset that can be used in any downstream NLP task, e.g. machine translation. RO-STS contains 212,619 tokens out of which 23,425 are unique. The average character length for the sentences is 66.39. The similarity scores for the sentences range from 0 to 5, with an average of 2.60. RO-STS is freely available in both the textual-similarity and the parallel corpus formats.[4]

To create the dataset, we first (i) obtained automatic translations using Google's translation engine. Then, (ii) the data was partitioned, checked, and corrected by 10 volunteers (ML researchers for whom Romanian is their native language and speak English fluently). These corrected partitions were then (iii) assigned to 3 volunteers (Romanian linguistic master students) for final validation. The volunteers in both phases (ii) and (iii) received the original English sentences and the Romanian translations from the previous phase with the instruction: "correct the translation if needed to make it sound like natural Romanian whilst keeping the meaning as close as possible to the original English version". Note that the sentences in the English STS dataset are short enough and quite clear content-wise, not leaving much space for interpretation or ambiguity for humans. However, the translation engine still makes errors (about 70% of the sentences were corrected in the first round),

---

[3]https://ixa2.si.ehu.eus/stswiki/index.php/STSbenchmark
[4]Available at: https://github.com/dumitrescustefan/RO-STS.

| Task | In-domain | Cross-domain |
|---|---|---|
| POS Tagging | 98.18 | 95.73 |
| Dependency Parsing | 90.38 | 88.97 |

| XQuAD-ro | F1 | EM |
|---|---|---|
| mBERT | 72.69 | 58.99 |
| XLM-R Large | 83.56 | 69.66 |

Table 2: Results for tasks on UD-RRT using the original in-domain splits and the proposed cross-domain splits.

Table 3: Zero-shot QA on XQuAD-ro.

but these errors (mainly erroneous idiom translation) can be easily spotted and corrected by native speakers. The second review round involved a smaller number of volunteers and their role was mainly to homogenize the translations, e.g. for English sentences where two different verb tenses could be used when translating in Romanian, annotators chose one and stuck to it. We provide here BLEU scores to give an idea about the volume of modifications made in the two phases: BLEU(Google translations, final) = 62.8. BLEU(first correction, final) = 77.9. This shows that the initial automatic translation was good, and the corrections made by volunteers improved the quality even more.

**XQuAD-ro.** The *XQuAD-ro* dataset contains the Romanian translations for the 240 paragraphs and 1,190 question-answer pairs of the XQuAD [1] dataset, previously available for 11 languages. We obtained the Romanian version with the help of professional human translators. The average number of tokens is 153.91 per paragraph, 12.03 per question and 3.33 per answer. The total number of tokens is 55,229, with 10,570 unique tokens. The average number of questions per paragraph is 4.95. The average character length of the paragraphs is 878.44, 67.01 for questions and 20.91 for answers. *XQuAD-ro* is already included in the official XQuAD repository for free public access.

**Wiki-ro.** The *Wiki-ro* dataset contains the July 2020 dump of the Romanian Wikipedia. It was thoroughly cleaned, with several custom rules. Besides removing all the wiki markup, we skipped wiki pages that have a large quantity of sequential numbers—there are many documents that are simply lists of years and events, unsuitable to calculate the perplexity of a language model. Other rules include limiting foreign words, punctuation, very short documents, proxy documents, etc. The corpus was segmented at the sentence level and tokenized, and is formatted as a one-sentence-per-line, with empty lines delimiting documents. The dataset is divided into train, validation, and test splits, always making sure that a document is entirely included in a single split. The train, validation, and test sets have 2.1M lines and 44M words, 14K lines and 276K words, and 16K lines and 327K words, respectively. The goal of this dataset is to provide standardized fine-tuning and evaluation of language modelling of Romanian text.

## 5 Experiments

### 5.1 Cross-genre splits for UD-RRT

UD-RRT [2] contains texts from 9 different genres: Academic, FrameNet, Journalistic, Law, Literature, Medical, Miscellanea, Science, and Wikipedia. The original dataset contains *within domain* train/valid/test splits for all 9 genres. Given the variability of the Romanian language across domains (caused by the use of specific vocabulary terms and phrases), we propose to use this dataset in a cross-domain setting for the tasks of dependency parsing and POS tagging, to better test the robustness of language models. To this end, we consider Miscellanea as the test domain, while using the remaining 8 domains for training. We chose Miscellanea as a test domain as it contains texts from all the other domains, plus some extra domains, e.g. dictionary definitions. This makes Miscellanea a good test set to probe generalisation, including to out-of-distribution samples. In Table 2, we can observe that the baselines are not sufficiently robust across domains, losing about 2% in accuracy on all the tasks. We used the Stanza framework [32] with default settings to run the in-domain and cross-domain experiments.

### 5.2 Cross-lingual Q&A baseline

For the XQuAD-ro dataset, we provide the same baseline results as the original XQuAD paper [1]. Namely, we use mBERT [13] and XLM-R Large [9] trained on the English SQuAD v1.1 training data and evaluate them via zero-shot transfer on XQuAD-ro test dataset. We report F1 and EM (exact match) in Table 3. XLM-R Large significantly outperforms mBERT. This is not surprising

| Model | Pearson coeff. | #params |
|---|---|---|
| RNN | 0.6853 | 15M |
| ro-BERT (cased) | 0.7927 | 124M |
| ro-BERT (uncased) | **0.8159** | 124M |
| mBERT (cased) | 0.7664 | 167M |
| mBERT (uncased) | 0.7690 | 167M |

Table 4: RO-STS baselines for semantic similarity.

| Training | WMT16 | RO-STS |
|---|---|---|
| RO-STS | 2.9 | 21.9 |
| WMT16 | 24.7 | 30.9 |
| WMT16 + RO-STS | **24.8** | 44.0 |
| RO-STS Finetuned | 24.6 | **45.9** |

Table 5: Translation results on WMT16 and RO-STS test sets.

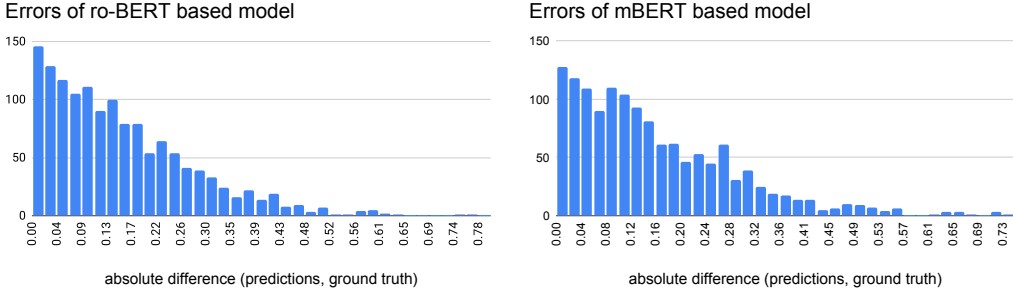

Figure 1: Errors made by two BERT-based models on the newly-created RO-STS dataset.

considering that the training set for XLM-R Large includes far more training data for Romanian than mBERT's training set: by volume of training data, Romanian is the 11th language for XLM-R Large and the 30th language for mBERT. In fact, out of the 12 languages present in XQuAD, XLM-R Large obtains the best results on Romanian, after English, in terms of both F1 and EM. The Russian influences present in Romanian and the fact that Russian is the second language by volume in XLM-R Large's training set might explain this performance.

### 5.3 RO-STS baselines

For RO-STS dataset, we provide baselines for two tasks: Romanian semantic textual similarity and EN → RO translation, given the parallel nature of the dataset.

**Semantic textual similarity.** We include three semantic similarity baselines: an RNN-based model and two transformer-based models, one using a monolingual Romanian BERT [ro-BERT; 15] and one using a multilingual BERT [mBERT; 13]. The RNN-based model uses a two-layer bidirectional LSTM to encode each sentence. Then, each sentence representation is passed through a standard additive attention layer. For the transformer models, we encode each sentence separately, then mean-pool the output token vectors. For all models, the similarity of the two resulting sentence representations is computed using the cosine distance. This similarity is then compared with the ground-truth scores normalized to $[0, 1]$. We use WordPiece tokenization and MSE loss for training. For the BERT-based models, we experimented with both the 'cased' and 'uncased' datasets.

The results of the three models are included in Table 4, together with their size. The RNN model is outperformed by the Transformer-based models in terms of Pearson coefficient. This is not surprising given that the RNN-based model was trained from scratch and has a much lower capacity. Figure 1 shows histograms of the errors made by the two transformer-based models. Note that ro-BERT is slightly more accurate than mBERT, the former having a more peaked histogram. In terms of normalized absolute similarity error, ro-BERT obtained 0.154 and mBERT 0.160.

**Translation.** We provide a baseline for RO-STS as a parallel corpus. We employ WMT16 RO-EN translation dataset [3] as a companion corpus and run the following experiments: (1) train on RO-STS, (2) train on WMT16, (3) train on both WMT16 and RO-STS, and (4) train on WMT16 and finetune on RO-STS. For modeling, we use the Open Neural Machine Translation (OpenNMT) toolkit [24] and we employ the original Transformer model [41]. The sentences were encoded using the Unigram subword tokenization [25], and we created a vocabulary of 8000 tokens for RO-STS training set and a vocabulary of 32000 tokens for the rest. The sentences were batched together by their approximate number of tokens resulting in batches of up to 2048 tokens for source and target sentences.

| Sentence pair (En translation provided for reference) | Sim | ro-BERT | mBERT |
|---|---|---|---|
| **Overestimating the similarity** | | | |
| *(Un bărbat dansează, Un bărbat şi o femeie dansează)* 
 In English: *(A man dances, A man and a woman dance)* | 0.4 | 0.76 | 0.80 |
| *(Un pisoi bea lapte dintr-un bol, Un copil mic bea apă dintr-o cană)* 
 *(A kitten drinks milk from a bowl, A small child drinks water from a cup)* | 0.16 | 0.69 | 0.50 |
| *(Nu ai nevoie de nicio viză, Nu ai nevoie de niciun fel de sos)* 
 *(You don't need any visa, You don't need any kind of sauce)* | 0 | 0.31 | 0.49 |
| **Underestimating the similarity** | | | |
| *(Te-ai prins, Ai înţeles bine)* 
 *(You got it, You understood well)* | 1 | 0.40 | 0.31 |
| *(Un bărbat râde cu o femeie, Un bărbat şi o femeie râzând)* 
 *(A man laughs with a woman, A man and a woman laughing)* | 0.96 | 0.37 | 0.44 |
| *(Eşti pe drumul cel bun, Ai perfectă dreptate)* 
 *(You are on the right track, You are perfectly right)* | 0.8 | 0.23 | 0.07 |

Table 6: Example of errors made by the baseline models in predicting the similarity of sentences from RO-STS test set. The 2nd column 'Sim' is the ground truth, with 0 meaning no relation between the sentence pair and 1 meaning perfectly similar. 3rd and 4th columns are ro-BERT's and mBERT's predictions.

We evaluate the models from our four settings on WMT16 and RO-STS test sets and measure their corresponding BLEU scores (see Table 5). The model trained on WMT16 obtains a BLEU score of 24.7 on WMT16 test and a BLEU score of 30.9 on RO-STS test. On the other hand, the model trained on RO-STS obtains a decent performance of 21.9 BLEU on RO-STS, but its performance is dramatically reduced on WMT16 test due to the small size of the training dataset and vocabulary, and domain mismatch. When training on both RO-STS and WMT16, the results on RO-STS were significantly improved by 13.1 BLEU, while the results on WMT16 were just slightly improved with 0.1 BLEU. The highest BLEU score on RO-STS was achieved by the model that was first trained on WMT16 and then finetuned on RO-STS, outperforming the previous model by 1.9 BLEU. However, as a result of fine-tuning on RO-STS, its performance slightly decreased on WMT16 by 0.1 BLEU.

## 5.4 Wiki-ro baseline

We run zero-shot evaluation with a pre-trained ro-BERT masked language model [15], calculating pseudo-loglikelihood scores (PLLs) and their corresponding pseudo-perplexities (PPPLs) as in [36], obtaining: 29.08 (P)PPL on the validation set and 28.00 (P)PPL on the test set. This is a first, modest baseline which could be significantly improved, e.g. by fine-tuning the model on Wiki-ro training set.

## 5.5 Gender debiasing baseline

We measure the gender bias in existing Romanian language embeddings [11] using the method proposed in [45] for languages with grammatical gender. The original paper measured the gender bias in Spanish and French, and proposed a mitigation method. We measure the gender bias for Romanian embeddings and provide this measure as a baseline to be improved by contributors. More specifically, we employ two sets $(A, B)$ containing paired words that define a semantic gender direction like *(tată, mamă)*, *(fiu, fiică)*[5]. We also employ two sets of (unpaired) frequently-used feminine and masculine nouns to define a grammatical gender direction. The correlation between these directions is higher in Romanian (0.53) compared to the reported value in [45] for Spanish (0.39). We project the grammatical gender component into the semantic gender direction to obtain orthogonal directions. To measure the gender bias, we consider two sets $(X, Y)$ of paired occupational embeddings, e.g. *(profesor, profesoară)*, *(inginer, ingineră)*[6], etc.; see Figure 2. Using the modified WEAT metric as in [45], we compute $b_w = ||s(w_m, A, B)| - s(w_f, A, B)||$, where $(w_m, w_f)$ are pairs in $(X, Y)$, and summing $b_w$ over the entire sets we get 2.57 (higher means more biased embeddings). This value is in between the values reported by [45] for Spanish and French (3.69 and 2.34, respectively). Note that it is difficult to make a direct comparison between these measures for different languages, but

---

[5]English *(father, mother), (son, daughter)*

[6]English (professor, engineer)

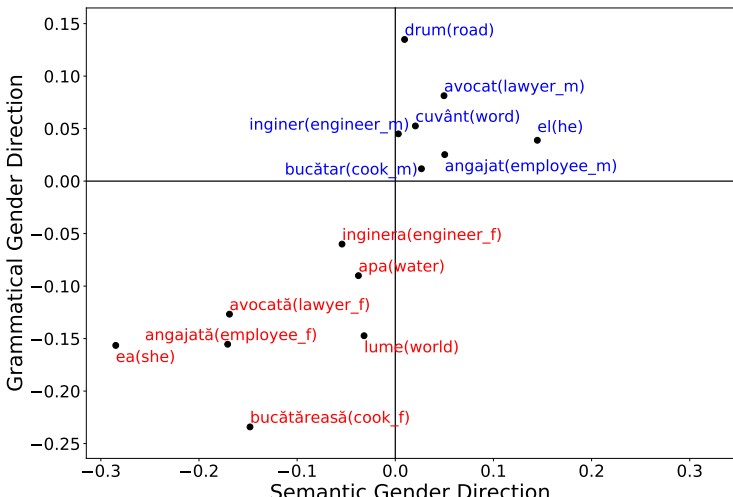

Figure 2: Occupational pairs and inanimate nouns projected on semantic gender direction (x-axis) and grammatical gender direction (y-axis). It can be observed that some feminine occupational words are farther away from the feminine definitional words than the masculine are from the masculine definitional words, revealing gender bias encoded in the embeddings.

we hypothesise that Romanian might contain more gender cues in the sentence structure and word inflexions due to its Slavic influences. Language models can pick up on these, leading to more biased embeddings. However, this requires a careful analysis to reach a clear conclusion. Our repository contains all the lists of words and a notebook to replicate this measurement.

## 5.6 Error analysis

The newly-created datasets allow analysing the errors made by the deep models, providing a useful glimpse into how much these models capture the semantics of the Romanian language.

**Semantic textual similarity.** We investigate the sentence pairs from the RO-STS test set where the models make large errors, i.e. they grossly overestimate or underestimate the similarity. We consider that the error is significantly large if the difference between the ground truth and the predicted similarity score is larger than an absolute value of 0.3. In this large error regime, we observe that both models have a tendency to overestimate the similarity of sentences: 10.7% pairs for ro-BERT and 10.6% pairs for mBERT. Moreover, mBERT has a slightly higher tendency to underestimate the similarity compared to ro-BERT: 3.2% pairs for mBERT compared to 2.6% for ro-BERT. At closer inspection, we observe that in most of the cases where the models overestimate the similarity, the sentence pairs have some parts in common, either the subject, the action, or the action's object. In this case, the models behave similarly to a bag-of-words model. The cases where the similarity is grossly underestimated contain idioms or the sentences have different word order. We include a few representative examples in Table 6.

**Machine translation.** We manually inspect the test samples with large errors and observe that in many cases the predicted translations are not identical, but are semantically similar to the ground truth; for example words replaced with their synonyms (e.g. Romanian: *acum → în prezent*, En: *now → in this moment*), adding or removing the article of a noun (e.g. Ro: *el cântă la chitară → el cântă la o chitară*, En: *He is playing guitar → He is playing the guitar*), or even paraphrasing entire chunks (e.g. Romanian: *Într-un sondaj realizat săptămâna trecută de CNN/ORC → Într-un sondaj CNN/ORC de săptămâna trecută*, English: *In a poll conducted last week by CNN/ORC → In a CNN/ORC poll of last week*). Such mistakes should not be penalized by a performance metric, as it is the case with BLEU. We believe that a dataset like RO-STS might prove useful for better estimating the quality of Romanian translations.

# 6 Conclusions

We proposed *LiRo*, a platform for benchmarking machine learning models across ten language understanding tasks for Romanian, with the explicit goal to increase accessibility and standardization, and to eventually accelerate progress. Additionally, we introduce, as part of *LiRo*, three new datasets: RO-STS, XQuAD-ro, and Wiki-ro. Wiki-ro is meant to provide a standardized evaluation dataset for language modelling. RO-STS and XQuAD-ro were obtained by human-translating their English counterparts and represent the first datasets of their kind for the Romanian language. We believe they play a dual role: first as standard benchmarks for Romanian semantic similarity and Q&A, respectively, allowing the evaluation of systems dedicated to these tasks. Second, as part of parallel corpora, they enable multilingual and cross-lingual research, which is of interest for the wider NLP community. *LiRo* also includes tasks on cross-domain splits of the standard UD-RRT dataset to test robustness of existing models and a task related to gender debiasing of Romanian language embeddings, to acknowledge the importance of this line of research and encourage works on Romanian embeddings debiasing. We pledge to continue extending *LiRo* by adding more tasks and datasets, either by creating them from scratch or, when possible, by translating existing datasets in other languages. Whilst the translation option comes with the risk of carrying over artifacts or errors from the original dataset, we believe that there is value in parallel corpora to enable cross-lingual tasks, which are of interest for the international NLP community.

## Acknowledgments

We thank the students from the West University of Timisoara, Department of Modern Languages and Literatures, enrolled in the 2020 Master's program, for contributing to the creation of RO-STS, and the Romanian Association for Artificial Intelligence and EEML for covering the translation fees for XQuaD-ro.

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
