# OpenReview forum: "LiRo: Benchmark and leaderboard for Romanian language tasks"
_NeurIPS.cc/2021/Track/Datasets_and_Benchmarks/Round1 — NeurIPS 2021 Datasets and Benchmarks Track (Round 1)_

### Official Review · Reviewer_adUi · 2021-07-02
**A good collection of benchmark tasks for Romanian NLP**

**Rating:** 8
**Confidence:** 4
**Clarity:** The paper is well written and the str…

**Strengths:**

A novel collection of datasets and tasks to promote research in NLP for Romanian language. The collection of tasks is well balanced and cover a lot of popular NLP tasks. The paper introduces three new datasets for Romanian that complement well the existing datasets. The website introduced and the resources on the website make it easy for a new researcher to get engaged in Romanian NLP.

**Weaknesses:**

The paper does not have many weaknesses. The only slight concern I have is with the RO-STS dataset which was created by using machine translation. Although the translations were corrected by human evaluators, it is hard to judge the quality of this dataset.

More generally I would have liked to see more information about the new datasets introduced.

Finally, as there has been a lot of discussion about annotation artefacts in some of the English NLU datasets, some discussion of this and how the datasets included here might or might not be subject to similar issues.

**Additional Feedback:**

A great addition to the field. I would like to see more of these benchmarks for different languages.

**Correctness:**

In general the datasets are constructed in a sound way. The only concern I have is with the RO-STS dataset as it was generated using machine translation (as discussed above).

**Documentation:**

Documentation is comprehensive

**Ethics:**

Potential biases in the newly introduced datasets would be good to mention in the paper.

**Relation To Prior Work:**

The related work section is detailed and mostly complete. As mentioned above, potential concerns related to annotation artefacts and other dataset biases would be good to include in the paper.

**Summary And Contributions:**

The paper introduces a new natural language processing benchmark for Romanian language. The benchmark include 10 tasks and introduces three completely new datasets. the tasks are well balanced, including semantic textual similarity, named entity recognition, questions answering as well as other popular NLP tasks. The paper introduces a homepage for the benchmark together with a leaderboard and other useful resources, like starter code to get started on using the benchmark. All in all this is a great addition to the field and will become a great baseline for creating similar benchmarks for other low resource languages.

---

> ### Author Response · Authors · 2021-07-14
> **Translation quality for Ro-STS and dataset artefacts**
>
> Thank you for your review. We agree with the concerns raised and we would like to provide some clarifications.
>
> Re: RO-STS translation: We will add more details in the paper: We started from an automatic translation that went through two rounds of human validation involving Romanian native speakers. The main reason why we think that this approach can lead to a good quality dataset is that the sentences in the English STS dataset are short enough and quite clear content-wise, not leaving much space for interpretation or ambiguity for humans. However, the translation engine still makes errors (about 70% of the sentences were corrected in the first round), but these errors (mainly erroneous idiom translation) can be easily spotted and corrected by native speakers. The second review round involved a smaller number of translators, whose role was mainly to homogenize the translations, e.g. for English sentences where two different verb tenses could be used when translating in Romanian, choose one and stick to it. We are confident that using a professional translator would not have resulted in a significantly different outcome.
>
> Re: annotation artefacts in English datasets: we agree that if there are inconsistencies in the original datasets, these are likely to be carried over in the translated dataset. As future work, we aim to perform more analysis of such artefacts. While creating Romanian datasets from scratch (which would alleviate this problem) is definitely on our agenda, we believe that there is value in parallel (translated) corpora for cross-lingual research carried out by the international NLP community.

---

### Official Review · Reviewer_SBUi · 2021-07-02
**good intuition; beneficial for NLP community**

**Rating:** 7
**Confidence:** 4
**Correctness:** To my knowledge, the statement is cor…

**Strengths:**

The benchmark consists of several datasets for Romanian language targeting many natural language tasks.

The authors intend to extend the benchmarks to non-NLP tasks like speech processing or image captioning.

**Weaknesses:**

It would be great if there are more discussion on the experimental results.

The authors did not explain why they chose the selected tasks in the benchmarks, but this is a minor issue. Also, are the baselines (i.e., NER, machine translation, sentiment analysis, etc) from prior work included in Table 1 the-state-of-the-art?

**Additional Feedback:**

As a side note, maybe the authors will be interested in making the benchmark dynamic, similar to Dynabench, to accelerate the progress for NLP.

**Clarity:**

It is well written. However, in line 298, do you have any explanations on why the correlation between gender direction is higher in Romanian than for Spanish?

**Documentation:**

N/A

**Relation To Prior Work:**

With the thorough discussion on related work, it is clear that the paper provides the first platform for Romanian NLP benchmarks.

**Summary And Contributions:**

The aim of this work is to increase the accessibility and standardisation for NLP community. The authors present benchmarks for 10 Romanian NLP tasks, along with three new datasets and baseline results on existing and newly created datasets. It appears convincing how the datasets are created. The experiments seem reasonable, accompanied with interesting analysis. Such leaderboard is very useful for widening multilingual/crosslingual language studies and evaluating language models.

---

> ### Author Response · Authors · 2021-07-14
> **Task selection and other clarifications**
>
> Thank you for your review.
> Re: more discussion of the experimental results: due to page limit, we had to keep the analysis short to be able to introduce the new datasets and baselines. If the paper is accepted, we will have one extra page that we intend to use for adding more analysis of the results.
>
> Re: tasks selection, we will expand on this in the paper: Criteria for selecting the tasks:
> -tasks that are generally considered as standard in the international community (e.g. text classification, POS tagging etc) to align with benchmarks in other languages,
> -tasks that have datasets publicly available and that we think have potential for practical impact: the goal of the benchmark is to enable research works, but also to facilitate the development of high-quality applications with high social impact for Romanian speakers.
> -cross-lingual tasks that allow bringing the Romanian language to the attention of international researchers working on cross-lingual studies
> -gender debiasing of embeddings task to encourage such studies for Romanian language.
>
> Regarding the results in table 1 extracted from the literature: to the best of our knowledge, these are SOTA results, we will clarify in the paper.
>
> Re: the difference in correlation of gender directions between Ro-Es, it is difficult to make a direct comparison: although Romanian and Spanish have common latin origin, Romanian has many influences from slavic languages as mentioned in the Introduction section. We hypothesise that one possible reason is that in Romanian there might be more gender cues in the sentence structure and word inflexions that a model can pick up on (https://scholar.harvard.edu/files/mpolinsky/files/Romanian_MIT_reprint.pdf). However, this requires a careful analysis to reach a clear conclusion.
>
> Re DynaBench: Thank you for the suggestion, we will consider it as future work.

---

### Official Review · Reviewer_7tKx · 2021-07-04
**A new NLP benchmark of 10 tasks on Romanian Language**

**Rating:** 7
**Confidence:** 3
**Clarity:** yes

**Strengths:**

The new benchmark has great value to the community in terms of low-resource language tasks. It has potential to increase accessibility and standardization of the research on Romanian language.

**Weaknesses:**

The performance of some tasks is already very high and it is unknown whether there will be significant method improvements based on this benchmark..

**Additional Feedback:**

n/a

**Correctness:**

The datasets are created in a standard way and the baselines are appropriate and reasonable.

**Documentation:**

The dataset is available and there are plenty of details in the paper on how the datasets are built.

**Relation To Prior Work:**

yes

**Summary And Contributions:**

1. This paper presents the first large-scale benchmark for Romanian language understanding. A couple of baselines, e.g., mBERT, XLR are tested.
2. In addition to Romanian language tasks, it also includes tasks that involves cross-lingual understanding.

---

> ### Author Response · Authors · 2021-07-14
> **High performance on some tasks**
>
> Thank you for your review. Regarding the high performance on some tasks, we agree that datasets that are close to saturation limit the model improvement that they can trigger. However, only 1 out of 10 tasks has accuracy around 95%, and 2 tasks are around 88%. We decided to still include these tasks as we hypothesise that it can be relatively easy to obtain performance close to 85% on these tasks using some simple LSTM-based models, but pushing accuracy to saturation (>98%) will require significant engineering, careful and extensive pre-training and fine-tuning. So they are still relevant for the Romanian NLP community, at least for improving the performance in practical applications.

---

### Decision · Program_Chairs · 2021-07-26

**Decision:**

Accept

**Comment:**

This paper presents a large-scale benchmark Romanian language with a website.
All reviewers make positive comments and believe that it would be a good contribution (useful for broadening multi-lingual language studies and the evaluation of pre-trained language models). So I recommend it is accepted.
As mentioned by one reviewer, more discussion could be made w.r.t experimental results.